# Counterfactual Techniques for Enhancing Customer Retention

## Abstract

In this paper, we introduce a novel method for generating counterfactual explanations to convert customers from an e-commerce company who frequently add items to their cart but do not proceed to checkout. We demonstrate that our method i) allows customization of mutable features, improving the practical applicability of our counterfactual explanations, and ii) outperforms existing techniques including DiCE, GANs, and CFRL in key metrics such as reconstruction error and L1 distance, while maintaining a low latency. Our method can easily be customized to optimize for high coverage or low latency by adjusting the number of nearest unlike neighbors, highlighting the trade-off between these competing goals.

## 1 Introduction

In various industries, counterfactual explanations can be used to analyze how a model's prediction would change if specific features were altered. For example, in e-commerce, customers often add items to their cart but do not proceed to checkout. Similarly, it is also important for loan companies Zhang et al. (2023); Grath et al. (2023), insurance providers Rye & Boyd (2022); Kumar & Ravi (Year), and fraud detection Whitrow et al. (2009); Ngai et al. (2011) companies to explain to their customers why they were denied loans or insurance. Understanding the reasons behind these behaviors and finding strategies to convert these customers is important for improving conversion rates. Recently, the demand for explainable AI has grown, becoming essential for increased transparency and trust among customers Gohel et al. (2023); Verma et al. (2020); Adadi & Berrada (2018).

Existing counterfactual explanations methods like Nearest Instance Counterfactual Explanations (NICE) Brughmans et al. (2022), Diverse Counterfactual Explanations (DiCE) Mothilal et al. (2019), Generative Adversarial Networks (GANs) Goodfellow et al. (2014), Counterfactual GANs (CounteRGAN) Nemirovsky et al. (2021), and Counterfactuals with Reinforcement Learning (CFRL) Samoilescu et al. (2021) often suffer from high latency, lack of customization (i.e., the user cannot specify which features are mutable), or suboptimal performance in terms of plausibility (i.e., how closely the counterfactual resembles the real world data) and distance (i.e., distance between the original instance and the counterfactual). Providing counterfactuals with actionable and customizable (i.e., mutable) features is essential for achieving business goals.

To address these challenges, we propose a novel counterfactual explanations method that supports customization of mutable features and allows the use of a variety of contextual embedding techniques to find the nearest neighbors from the opposite class. Our approach involves converting each data row into text, considering both feature and value, and then generating embeddings using a BERT Devlin et al. (2018) model. In the e-commerce domain, we specifically use eBERT, which is fine-tuned on e-commerce product titles. Through BERT-based embeddings, feature values are represented semantically, making it an effective method for finding neighbors that are similar but have a different predicted label.

Our new technique is particularly valuable in production systems because it supports customization, allowing business users to specify which features can be changed. As a result, the generated counterfactuals are more actionable. Compared to baseline methods that support customization, our method performs best in coverage, reconstruction error, and L1 distance.

## 1.1 Contributions

The primary objective of this research is to develop a novel counterfactual explanations method to enhance customer retention in e-commerce settings. Our method aims to provide explainable, actionable insights that help identify key factors influencing customers' decision-making processes, specifically for scenarios where customers abandon their carts before completing a purchase. We have achieved the following contributions:

- Developed a new counterfactual explanations algorithm that improves upon current state-of-the-art methods such as NICE, DiCE, and GAN-based approaches by balancing coverage, latency, and plausibility of counterfactuals.

- Introduced an embedding-based approach using BERT to generate highly plausible counterfactuals that more accurately reflect customer behavior in an e-commerce setting.

- Ensured that the proposed method supports customization of mutable features, allowing business stakeholders to specify which factors can be realistically adjusted to achieve desired outcomes, thus enhancing the practical applicability of counterfactual explanations.

- Evaluated the effectiveness of the proposed method against existing techniques using key metrics such as coverage, reconstruction error, and L1 distance, and demonstrated its applicability in real-world datasets, including both a proprietary e-commerce dataset as well as the public adult income benchmark.

This work aims to address current limitations in counterfactual explanations, offering a comprehensive and actionable solution for improving customer retention.

## 2 Data Preprocessing

### 2.1 E-commerce

Our proprietary dataset consists of 200,000 shopping sessions to understand customer behavior. The dataset includes a total of 47 features extracted primarily from User, Cart, and Listing tables. Of these features, 36 are categorical, such as product categories and purchase data, while the remaining 11 are numerical variables, including shipping costs, item prices, and feedback scores. The target value is the outcome of the shopping session, either 1 (i.e., success) if the customer successfully checked out or 0 (i.e., failure) otherwise.

To preprocess the data for embedding generation, we scaled the price and shipping fee features between 0 and 100 and categorized them into four buckets. The price features were divided into four buckets: 0-20 (*budget*), 20-40 (*affordable*), 40-60 (*premium*), and 60-100 (*luxury*). Shipping fees were also categorized into four buckets: 0-25 (*low*), 25-50 (*medium*), 50-75 (*high*), and 75-100 (*very high*). In an ablation study, we compared to embeddings generated without bucketization.

After bucketizing, we removed null values, scaled numerical feature values using a standard scaler, and encoded categorical data using a binary encoder. We explored several encoding methods for the categorical features, including label encoding, one-hot encoding, target encoding, and binary encoding. While one-hot encoding achieved slightly higher accuracy, it led to overfitting due to high cardinality, increasing the dataset to over 9,000 features and significantly increasing computation time. Binary encoding, although slightly less accurate, reduced dimensionality to 165 features, preventing overfitting and significantly reducing the computation time. Binary encoding was chosen due to its ability to reduce dimensionality and computation time, while preventing overfitting.

The dataset was divided into a training set of 160,000 instances and a test set of 40,000 instances to evaluate model performance. This preprocessing step ensured that both categorical and numerical features were ready for embedding generation.

### 2.2 Adult Income

The public adult income dataset consists of 32,561 adults, split into 80% training and 20% testing, with a target variable indicating whether or not the individual's income exceeds $50,000 per year.

Table 1: Performance metrics of various classifiers on an e-commerce binary classification task for whether or not a customer completed a purchase.

|                     | Accuracy | Precision | Recall | F1   |
|---------------------|----------|-----------|--------|------|
| Random Forest       | **89.0** | **88.7**  | 89.3   | **89.0** |
| MLP                 | 82.1     | 82.4      | 82.1   | 82.1 |
| Logistic Regression | 66.4     | 66.4      | 66.4   | 66.4 |
| XGBoost             | 84.7     | 81.4      | **89.7** | 85.4 |

There are 14 features, including categorical features such as workclass, race, and sex, as well as numerical features such as age, hours per week, capital gain, and capital loss.

Just as we did with the e-commerce dataset, we preprocessed the data for embedding generation by encoding the numerical features with a standard scaler and categorical features with a binary encoder prior to converting feature and values to text. We also investigated the effects of bucketizing age into three buckets for ages 0-30 (*young*), 30-50 (*middle-aged*), and 50-100 (*elderly*) versus bucketizing age into eight finer-grained buckets, one for each decade, as well as bucketizing hours per week into three buckets: 0-20 hours (*part-time*), 20-40 (*full-time*), and 40-100 (*overtime*).

## 3 CLASSIFICATION

Counterfactual explanations algorithms incorporate classifiers by using their predictions as feedback during the generation process. Each counterfactual is tested to see if it successfully flips the classifier's original decision or not. We implemented four different types of classifiers—a Random Forest (RF), Logistic Regression, XGBoost Chen & Guestrin (2016), and Multilayer Perceptron (MLP).

As shown in Table 1, the RF classifier showed promising results and was selected for generating counterfactual explanations. With the highest F1 score of 89% on the held-out e-commerce dataset, the RF classifier demonstrated its ability to properly classify and understand customer behavior. The Logistic Regression classifier performed the worst.

## 4 BASELINE COUNTERFACTUAL METHODS

Here, we describe four strong baseline counterfactual explanations methods we compared against.

*Diverse Counterfactual Explanations (DiCE)* Mothilal et al. (2019) primarily focuses on generating feasible and diverse counterfactuals. It extends counterfactual explanations by incorporating determinantal point processes (DPP) Kulesza et al. (2012), which is a probabilistic model used to ensure diversity in the generated examples. This allows DiCE to provide a range of alternatives for changing outcomes. DPP selects a subset of diverse examples by maximizing the determinant of a kernel matrix built from the examples. This diversity is balanced against proximity, which measures the closeness of counterfactuals to the original input. The method optimizes a loss function that combines y-loss (the difference in prediction), proximity, and diversity, adjusted using hyperparameters $\lambda_1$ and $\lambda_2$. The counterfactuals are generated through gradient descent, which iteratively adjusts feature values to meet the objective while respecting any real-world feature constraints.

While DiCE focuses on generating diverse counterfactuals, it suffers from lower plausibility, i.e., high reconstruction error and L1 distance. In contrast, our method is customizable while still achieving low reconstruction error and L1 distance, providing more actionable and plausible counterfactuals, especially in e-commerce applications where proximity is more critical than diversity.

*Nearest Instance Counterfactual Explanations (NICE)* Brughmans et al. (2022) generates counterfactual explanations using a nearest unlike neighbor-based approach. The algorithm identifies the nearest neighbor with a different class label and changes one feature value at a time from the original instance to match that of the neighbor. This process generates hybrid instances, guided by a reward function that prioritizes sparsity, proximity, or plausibility, depending on the specific NICE variant.

Although NICE produces counterfactuals with minimal L1 distance, it lacks support for customization of feature mutability, limiting its practical applicability, especially in the e-commerce domain.

*Generative Adversarial Networks (GANs)* CounteRGAN Nemirovsky et al. (2021) is an extension of Residual GAN (RGAN), designed to generate realistic and actionable counterfactuals by applying small perturbations to existing data points rather than creating new instances from scratch. The idea is to generate subtle modifications that can flip a model's prediction, while ensuring that the changes are realistic and feasible. In RGAN, the generator produces perturbations that modify input data, while the discriminator attempts to distinguish between real and modified data. CounteRGAN adds a target classifier to this process, which ensures that the generated counterfactuals not only resemble real data but also result in the desired predicted label change.

Standard GAN and CounteRGAN generate counterfactuals with the lowest latency, but at the cost of higher L1 distance, which reduces plausibility. Our method, while slightly slower, strikes a balance by providing highly plausible counterfactuals with lower L1 distance, making it more effective in producing realistic and actionable outcomes.

*Counterfactuals using Reinforcement Learning (CFRL)* Samoilescu et al. (2021) uses a reinforcement learning framework for counterfactual generation, transforming the optimization process into a learnable task. It enables the generation of multiple counterfactuals in a single forward pass, relying solely on the feedback from the classifier's predictions. This model-agnostic method allows for feature-level constraints, ensuring real-world feasibility. CFRL uses a critic to estimate rewards from the environment and an actor to output counterfactual latent representations. This method enables high flexibility, as feature-level constraints like immutability can be incorporated via conditioning vectors, ensuring that the generated counterfactuals are plausible and actionable.

The documentation for CFRL says that it supports mutable feature customization. However, in our experiments, the generated counterfactuals did not respect the mutable feature constraints, and it had a larger L1 distance compared to our method. By offering a better balance between feature mutability, coverage, and proximity, our method produces more actionable and realistic counterfactuals.

Other counterfactual generation methods such as CERT Sharma et al. (2020) and MACE Karimi et al. (2020) were not considered due to their high latency Schleich et al. (2023). GeCo Schleich et al. (2023), although it offers many customization options for the counterfactual, suffers from very low coverage Brughmans et al. (2022).

## 5 OUR METHOD

Our novel embedding-based counterfactual explanations method is designed to address the limitations of NICE, GANs, and CFRL by supporting customization of feature mutability. In addition, our method provides support for generating BERT-based embeddings that capture deep semantic relationships within the data, in order to identify more plausible and actionable counterfactuals.

### 5.1 EMBEDDING GENERATION USING BERT

In our method, each data sample is transformed into a text representation to generate embeddings for counterfactual explanations. This transformation involves converting all feature names and their respective values into a textual format. For example, a data sample with the following attributes:

- `PRICE = ` *affordable*
- `SHPNG_COST = ` *low*
- `PAYMNT_TYPE = ` *CreditCard*
- `QTY_SOLD = ` *2*

would be represented as: "PRICE *affordable* SHPNG_COST *low* PAYMNT_TYPE *CreditCard* QTY_SOLD *2*." This text is then standardized to lowercase to maintain consistency before pre-processing. The preprocessed text for each of the features in each dataset is input into either SentenceBERT Reimers & Gurevych (2019) or the e-commerce-specific eBERT model. The eBERT model outputs a 768-dimensional embedding for each data sample, capturing the semantic relationships of the data. In our ablation study, we compare these BERT models to no embeddings (i.e.,

simply representing each data sample as a vector of all its feature values), using the embedding of an autoencoder trained on the dataset, and using an embedding generated by TabTransformer Huang et al. (2020) for tabular datasets.

These vector representations serve as the basis for the next step in our method, which involves identifying the nearest unlike neighbors. This process is crucial for generating plausible and contextually relevant counterfactual explanations by calculating the distances between the generated embeddings.

## 5.2 NEAREST UNLIKE NEIGHBORS GENERATION

After generating vector representations of the data, the next step involves finding the nearest unlike neighbors. Our method employs FAISS (Facebook AI Similarity Search) Facebook Engineering (2017) IndexFlatL2 to identify the $k$ nearest unlike neighbors based on a similarity metric (i.e., L2 distance or inner product). FAISS is a library optimized for fast similarity searches, particularly for high-dimensional vectors such as embeddings. We compare two index types (i.e., IndexFlatL2 and IndexFlatIP) for searching for nearest neighbors, using either L2 or inner product distances. For example, a vector representing an unsuccessful shopping session may be queried against the index containing vectors of successful sessions to find the $k$ nearest neighbors with the "success" label.

The nearest unlike neighbors retrieved represent instances that lie on the opposite side of the decision boundary. Each of these neighbors has different values for features such as PRICE, SHPNG_COST, and PAYMNT_TYPE, which makes them candidates for generating counterfactuals. The overall process involves building the FAISS index, adding all dataset vectors, associating each vector with a class label, and then retrieving the $k$ nearest unlike neighbors using FAISS.

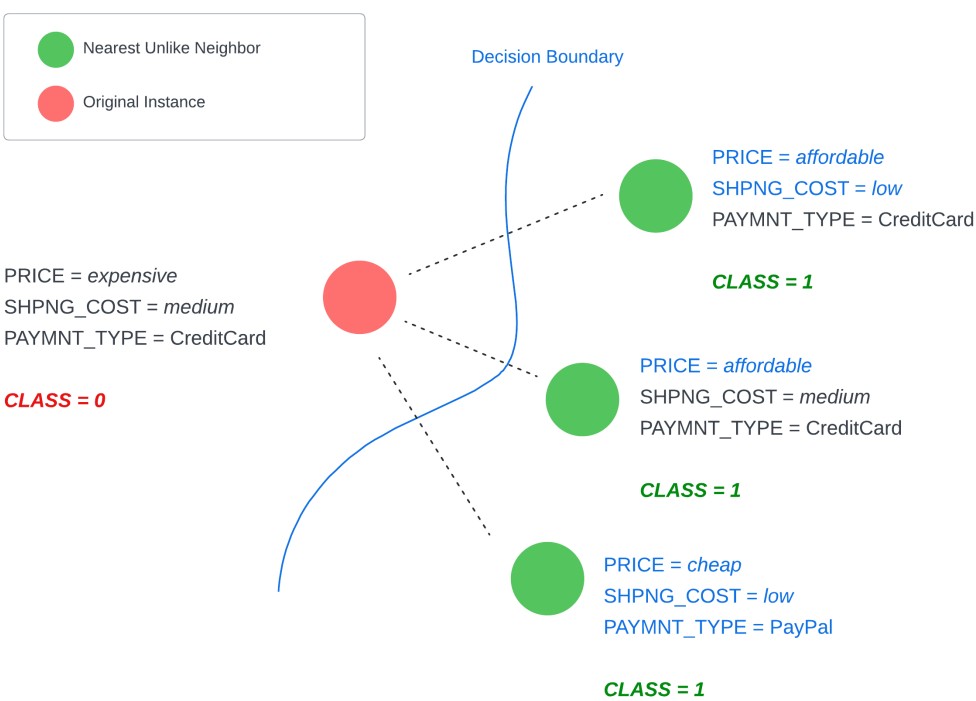

Figure 1: The original instance (red) has a PRICE of *expensive*, a SHPNG_COST of *medium*, and uses PAYMNT_TYPE = *CreditCard*. The class is 0, indicating that this customer is not predicted to make a purchase. The nearest unlike neighbors (green) are counterfactual instances with slightly different features, located on the opposite side of the decision boundary (i.e., the class is 1, and thus are predicted to make a purchase).

Figure 1 illustrates the concept of nearest unlike neighbors in relation to the decision boundary. The original instance (red dot) is positioned near several nearest unlike neighbors (green dots) that lie on the opposite side of the decision boundary. This proximity allows for generating counterfactuals by modifying features in a way that crosses the decision boundary, achieving a different classification.

## 5.3 Counterfactual Search

Once we identify the nearest unlike neighbors, the next step is to generate counterfactuals using a greedy heuristic search method. In counterfactual search, we find data points that can be used to alter a classifier's decision, while making sure that they have a low distance from the original instance.

Using the neighbors identified in the previous section, the greedy heuristic search modifies features one at a time. As shown in Figure 1, the original instance has a PRICE of *expensive*, a SHPNG_COST of *medium*, and uses PAYMNT_TYPE = *CreditCard*. We modify each of these features individually to match one of the nearest unlike neighbors. For example, lowering the PRICE from *expensive* to *affordable* or lowering the SHPNG_COST from *medium* to *low* could flip the prediction from class 0 to class 1.

As the features are adjusted, the heuristic search keeps track of how each modification affects the classifier's decision. The goal is to generate the closest counterfactual, i.e., the one that modifies the fewest features while successfully flipping the class. In some cases, a combination of modifications, like changing both PRICE and SHPNG_COST, may be required.

The Counterfactual Generation Algorithm 1 seeks to identify the closest counterfactual instance to an original data point by iteratively modifying its features to achieve a desired target classification. Initialized with the original instance $x_{\text{orig}}$, the algorithm progresses through the feature space by comparing against a set of nearest unlike neighbors $\mathbb{N}$. For each neighbor, the algorithm creates $X_{\text{mod}}$, which is a set of new instances derived from $x_{\text{orig}}$, where each instance in $X_{\text{mod}}$ is generated by altering one specific mutable feature of $x_{\text{orig}}$ to match the corresponding feature in the neighbor.

---

**Algorithm 1** Counterfactual Generation

---

**Require:**
  $M$: Prediction model
  $E$: Encoders for categorical features
  $S$: Scaler for numerical features
  $\mathbb{N}$: Set of nearest unlike neighbors

  **Initialize:**
  $x_{\text{cf}} \leftarrow x_{\text{orig}}$           ▷ Initialize counterfactual candidate with original instance
  $d_{\min} \leftarrow \infty$           ▷ Initialize minimum distance
**Ensure:**
  $x^*$: Optimal counterfactual instance
  $d_{\min}$: Minimum distance
1: **for** $n \in \mathbb{N}$ **do**
2:      **for** $t \leftarrow 1$ to $Iter_{\max}$ **do**
3:          $X_{\text{mod}} \leftarrow \{x' : x' \text{ varies from } x_{\text{cf}} \text{ by one feature towards } n\}$    ▷ Generate new instances
4:          $(x', d) \leftarrow \text{EvaluateModifications}(X_{\text{mod}}, x_{\text{cf}}, E, S, M)$    ▷ Evaluate counterfactuals
5:          **if** $d < d_{\min}$ **then**
6:             $d_{\min} \leftarrow d$
7:             $x^* \leftarrow x'$           ▷ Update optimal counterfactual
8:             $x_{\text{cf}} \leftarrow x'$           ▷ Update current counterfactual
9:          **end if**
10:         **if** $M(x_{\text{cf}}) = \text{target class}$ **then**
11:            **return** $(x_{\text{cf}}, d_{\min})$          ▷ Return if target class is achieved
12:         **end if**
13:      **end for**
14: **end for**
15: **return** $(x^*, d_{\min})$

---

Each modified instance, denoted as $x'$, is evaluated for its potential as a counterfactual: the modifications are checked to ensure they not only bring $x'$ closer to achieving the target classification but also maintain minimal distance from $x_{\text{orig}}$. This iterative process continues until a satisfactory counterfactual that meets the target classification is found, or all possibilities are exhausted. This method ensures that each proposed counterfactual is a minimal and interpretable adjustment to the instance that alters the model's decision.

# 6    EXPERIMENTS

We conducted experiments on a randomly held-out test set of 1,000 data points using all the counterfactual techniques in Section 4 and compared them on both datasets using the following metrics:

- *Reconstruction Error:* Measures how closely a counterfactual instance resembles real-world data. It is calculated as the L2 norm between the counterfactual instance and the output of an autoencoder trained to predict the original training datapoints, given noisy training data as input. A lower reconstruction error indicates a more plausible and realistic counterfactual Looveren & Klaise (2019); Dhurandhar et al. (2018); Nemirovsky et al. (2021). Our autoencoder consists of two 32-dimensional fully-connected layers with tanh activation and a final layer with tanh activation. This error is defined as

$$E = \|AE(x_{cf}) - x_{cf}\|_2^2 \tag{1}$$

  where $AE$ is the autoencoder model and $x_{cf}$ is the counterfactual instance.

- *L1 Distance:* Measures the distance between the original and counterfactual instances. Again, a lower L1 distance indicates a more plausible and realistic counterfactual.

$$L1(x_{orig}, x_{cf}) = \sum_{i=1}^{d} |x_{orig,i} - x_{cf,i}| \tag{2}$$

- *Latency:* Estimates the time (in seconds) required to generate a single counterfactual.

- *Coverage:* Represents the proportion of test set instances for which counterfactuals can be successfully found, calculated as

$$C = \frac{\sum_{i=1}^{n} \mathbf{1}\left(M(x_{cf}^{(i)}) = d\right)}{n} \times 100 \tag{3}$$

  where $n$ is the total number of instances, $M$ is the prediction model, $d$ is the desired class (i.e., the opposite of the predicted class), and $\mathbf{1}(\cdot)$ is the indicator function that equals one if its argument is true and zero otherwise. Coverage calculates the percentage of instances where the model's prediction for the counterfactual is the desired class.

The results of our evaluation on the held-out test set for both datasets, where all features are considered mutable, are shown in Tables 2 and 3. We see that our method has the lowest reconstruction error compared to all baselines on both datasets. However, NICE has the lowest L1 distance, and the GANs have the lowest latency. On the e-commerce dataset, our method suffers from high latency due to the higher dimensionality of the data. Although NICE appears to be the best method overall when using all feature values, it does not allow customization of mutable features and thus is impractical for real-world business use cases, which is the primary advantage of our approach.

Our experiments in tuning hyperparameter $k$ (i.e., the number of nearest unlike neighbors) on the e-commerce dataset show that increasing $k$ results in higher coverage, at the expense of higher latency. Our experiments with different bucketization strategies and distance metrics on the e-commerce dataset show that L2 distance slightly outperforms inner product, and that using a few buckets outperforms both using more fine-grained buckets and not bucketizing before generating embeddings.

In Table 4, we see the results of experiments on both datasets with only one mutable feature (i.e., current price for the e-commerce dataset and hours per week for the adult income dataset). Here is where our method shines—it outperforms DiCE in coverage, reconstruction error, and L1 distance.

Table 2: A comparison of all the counterfactual methods on four evaluation metrics on the e-commerce dataset. We vary hyperparameter $k$ to demonstrate the tradeoff between coverage and latency. Note that the $\pm$ values represent the 95% confidence interval, calculated as $(1.96 \times \frac{\sigma}{\sqrt{n}})$, where $\sigma$ is the standard deviation and $n$ is the sample size.

| Method | Coverage | Reconstruction Error | L1 Distance | Latency (s) |
|--------|----------|----------------------|-------------|-------------|
| DiCE | 100% | 52.0 ± 27.6 | 131 ± 52.0 | 1.82 ± 0.07 |
| NICE | 100% | 7.03 ± 0.31 | **1.27 ± 0.61** | 0.32 ± 0.01 |
| CFRL | 100% | 6.14 ± 0.007 | 56.0 ± 0.54 | 1.28 ± 0.009 |
| Standard GAN | 100% | 7.26 ± 0.00 | 70.2 ± 0.40 | **0.01 ± 0.002** |
| CounteRGAN | 100% | 7.08 ± 0.09 | 32.1 ± 0.33 | 0.06 ± 0.001 |
| Ours ($k = 12$) | 99.5% | 3.75 ± 0.11 | 41.4 ± 0.75 | 2.49 ± 0.10 |
| Ours ($k = 24$) | 99.8% | 3.75 ± 0.11 | 34.5 ± 1.01 | 2.46 ± 0.10 |
| Ours ($k = 36$) | 99.9% | 3.75 ± 0.11 | 34.4 ± 1.10 | 2.50 ± 0.10 |
| Ours ($k = 48$) | 100% | **3.75 ± 0.11** | 19.8 ± 1.12 | 2.42 ± 0.10 |

Table 3: A comparison of all the counterfactual methods on four evaluation metrics on the adult income dataset. Our method uses $k = 12$ for each experiment, only varying the bucketization strategy or distance metric used to find nearest unlike neighbors. The default is three age buckets, whereas with fine-grained age buckets, there are eight buckets, one for each decade.

| Method | Coverage | Reconstruction Error | L1 Distance | Latency (s) |
|--------|----------|----------------------|-------------|-------------|
| DiCE | 100% | 5.93 ± 0.23 | 7.82 ± 0.28 | 0.29 ± 0.003 |
| NICE | 100% | 1.27 ± 0.08 | **2.62 ± 0.152** | 0.10 ± 0.002 |
| CFRL | 100% | 4.57 ± 0.02 | 20.5 ± 0.20 | 0.13 ± 0.001 |
| Standard GAN | 100% | 3.55 ± 0.06 | 14.5 ± 0.11 | **0.06 ± 0.001** |
| CounteRGAN | 100% | 3.77 ± 0.06 | 16.5 ± 0.11 | 0.07 ± 0.001 |
| Ours | 100% | **1.20 ± 0.08** | 4.70 ± 0.24 | 0.15 ± 0.004 |
| Ours (*no buckets*) | 100% | 1.24 ± 0.08 | 4.62 ± 0.23 | 0.14 ± 0.003 |
| Ours (*8 age buckets*) | 100% | 1.24 ± 0.07 | 5.00 ± 0.24 | 0.15 ± 0.004 |
| Ours (*inner product*) | 100% | 1.22 ± 0.05 | 6.75 ± 0.31 | 0.20 ± 0.005 |

However, our method has higher latency, just as when all features are mutable, but in this more realistic use case, the latency is less than one second per counterfactual generated, for both datasets. While CFRL's documentation claims that it allows the user to set mutable features, it does not in practice always respect these constraints since it violated them for every instance in both datasets.

In our ablation study, we see that BERT-based embeddings outperform all other embedding types, with eBERT best for the e-commerce dataset and SentenceBERT best for the adult income dataset. This makes sense intuitively since BERT trained on e-commerce data is likely to have learned the semantic relationships among words used in product titles better than SentenceBERT, and vice versa.

In Table 5, we see the same pattern as with only one mutable feature for two, three, and four mutable features, where our method is best in terms of coverage, reconstruction error, and L1 distance, but worse than DiCE in terms of latency. Again, performance is higher overall for the adult income dataset than the e-commerce dataset. Perhaps related to this, we see a bigger gap in coverage between our method and the baseline method on the e-commerce dataset. Again, if we increase $k$, the improvement in performance of our method compared to the baseline also increases.

Finally, we performed feature importance analysis to determine which features are most important in the predictions made by the random forest classifier and, thus, most important in flipping the prediction to the opposite class during counterfactual generation. With the feature importance assigned to each of the features by the random forest classifier itself, the most important features for the adult income dataset are work class and age, which are two of the four mutable features we used. Using SHAP, the most important features for the adult income dataset are marital status, education level,

Table 4: A comparison of DiCE to our method on both datasets when only one feature is mutable. We set $k = 12$ and vary the embeddings, e.g., raw feature values (*no embeds*) vs TabTransformer.

| Method | E-commerce Dataset | | | | Adult Income Dataset | | | |
|---|---|---|---|---|---|---|---|---|
| | Coverage | Error | L1 | Latency | Coverage | Error | L1 | Latency |
| DiCE | 4.8% | 26.3 ± 7.33 | 67.4 ± 9.39 | **0.45 ± 0.05** | 23.3% | 1.60 ± 0.13 | 13.2 ± 0.51 | **0.15 ± 0.003** |
| Ours (*no embeds*) | 6.4% | 3.67 ± 0.15 | 43.6 ± 2.20 | 0.84 ± 0.02 | 22.4% | 1.09 ± 0.11 | **12.4 ± 0.58** | 0.20 ± 0.006 |
| Ours (*TabTransf*) | 6.1% | 3.76 ± 0.17 | 43.0 ± 2.58 | 0.86 ± 0.02 | 24.9% | 1.10 ± 0.10 | 12.4 ± 0.53 | 0.22 ± 0.006 |
| Ours (*autoencoder*) | 7.5% | **3.65 ± 0.14** | **41.7 ± 2.34** | 0.85 ± 0.02 | 23.8% | **1.06 ± 0.10** | 12.6 ± 0.51 | 0.22 ± 0.006 |
| Ours (*eBERT*) | **11.5%** | 3.72 ± 0.20 | 43.1 ± 1.80 | 0.90 ± 0.02 | 24.7% | 1.21 ± 0.11 | 12.8 ± 0.53 | 0.22 ± 0.007 |
| Ours (*SentBERT*) | 11.1% | 3.69 ± 0.20 | 43.1 ± 1.66 | 0.86 ± 0.02 | **26.2%** | 1.15 ± 0.10 | 12.7 ± 0.51 | 0.23 ± 0.006 |

Table 5: Coverage of DiCE vs our method for two, three, or four mutable features. For e-commerce, the second mutable feature is shipping cost, third is payment type, and fourth is unit price. For adult income, the second mutable feature is occupation, third is work class, and fourth is age.

| Method | E-commerce Dataset | | | Adult Income Dataset | | |
|---|---|---|---|---|---|---|
| | 2 features | 3 features | 4 features | 2 features | 3 features | 4 features |
| DiCE | 3.9% | 7.8% | 12.3% | 27.1% | 30.2% | 41.8% |
| Ours (*eBERT*) | **15.7%** | **15.4%** | **23.9%** | **28.3%** | **31%** | **44.5%** |

capital gain, age, hours per week, relationship, capital loss, and occupation; again, age is important, as well as the other features we considered mutable, including hours worked per week and occupation. The small bump in coverage when we added the second and third features, occupation and work class, to the set of mutable features could be due to the lack of importance for these two features relative to age and hours per week. For the e-commerce dataset, current price and unit price are third and fourth most important for the RF classifier and fifth and sixth most important according to SHAP. With the RF feature importance, shipping cost ranks ninth out of 47 features, and payment type is lower, at only 27 out of 47. Again, this is reflected in the lack of improvement in coverage when we add the third feature, payment type, to the set of mutable features.

## 7 CONCLUSION

In this paper, we introduced a novel counterfactual explanations method that uses embeddings generated by a BERT model to create more accurate and actionable counterfactuals than existing approaches and, more importantly, our technique supports customization of mutable features. This is especially important in e-commerce settings where companies wish to determine how best to adjust specific features such as price in order to nudge a customer to complete a purchase.

Our experiments conducted on 200K shopping sessions and the public adult income benchmark demonstrate that our method outperforms existing counterfactual generation methods in terms of coverage, reconstruction error, and L1 distance, with a tradeoff between plausibility and latency. The final latency of 0.3 seconds per counterfactual on the adult income dataset with three mutable features indicates that our method is suitable for real-time applications. In future work, we aim to further reduce latency and add functionality to specify a range of values for each mutable feature, which will improve the customization and real-world usability of our method.

## 8 ETHICS STATEMENT

The business use case of generating counterfactuals raises ethical concerns about consumer manipulation, the potentially unfair treatment of certain customer groups, and lack of transparency to customers. To address fairness and privacy concerns, the e-commerce dataset is anonymized, and no personal information is used, other than whether they are verified as an adult, whether they take

part in surveys, whether they are a registered bulk user, and whether they want customer support, direct mail, or telemarketing. The majority of the 47 features are all related to the product in the cart, including its category ID, price information, quantity, shipping, and seller rating, etc.

# 9    THE USE OF LARGE LANGUAGE MODELS (LLMs)

LLMs were used to polish the writing of some sections of this paper.

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
