# OpenReview forum: "Counterfactual Techniques for Enhancing Customer Retention"
_ICLR.cc/2026/Conference — Submitted to ICLR 2026_

### Official Review · Reviewer_gEno · 2025-10-16

**Soundness:** 2
**Presentation:** 2
**Contribution:** 2
**Rating:** 4
**Confidence:** 4

**Summary:**

The paper proposes a BERT-embedding-based approach to generate counterfactual explanations for e-commerce user behavior, specifically targeting users who add items to their carts but do not check out. The method converts tabular data into text, embeds each instance using BERT or eBERT, retrieves nearest unlike neighbors with FAISS, and applies a greedy heuristic to produce actionable counterfactuals while respecting feature mutability constraints.

**Strengths:**

The practical focus on e-commerce user retention is relevant for applied XAI.
The experiments compare to a reasonable set of baselines and report multiple metrics.
The inclusion of both a proprietary and a public dataset adds some external validity.

**Weaknesses:**

The author mentioned two issues which are suboptimal performance in terms of plausibility (i.e., how closely the counterfactual resembles the real world data) and distance (i.e., distance between the original instance and the counterfactual). I got confused about these two issues, can author give a simple example to explain them? why we need to address them?

The method essentially merges BERT embeddings with nearest-neighbor search; is there any technical improvement?

Because this work is more about real-world e-commernce senario, and the approach depends on heavy embedding computation for every instance, so are there any scaling issues?

Authos claimed they improved latency, but latency remains higher than baselines?

**Questions:**

Please refer to the weakness part

---

### Official Review · Reviewer_iFeW · 2025-10-29

**Soundness:** 2
**Presentation:** 1
**Contribution:** 1
**Rating:** 2
**Confidence:** 4

**Summary:**

This paper introduces a method for generating counterfactual explanations (CFEs) in the context of e-commerce—in particular, centered on the setting where customers (users) add items to their cart but do not complete checkout. The authors evaluate their method against existing methods (DiCE, NICE, Counterfactuals with RL, and two GAN methods) on two datasets: (1) e-commerce and (2) UCI Adult Income and report performance in terms of coverage (proportion of test set instances for which counterfactuals can be successfully found), reconstruction error (L2 norm), L1 distance, and latency (time in seconds to generate a single counterfactual).

**Strengths:**

In terms of the empirical evaluation presented:
* Performance metrics reported are appropriate for the task (coverage, L2 norm, L1 distance, and compute time).
* In the setting of algorithmic recourse, feature importance analysis as done here can be useful for interpretability and helping identify the most impactful features a user can adjust to increase the likelihood of changing their outcome for a particular task.

**Weaknesses:**

* The setting of the work is very narrow. While an application area can be useful for grounding the problem and informing methodological desiderata, it is unclear how useful this approach is even in the context of e-commerce, let alone other related or disparate areas where counterfactual explanations are useful.
* The experimental evaluation is very limited. While UCI Adult dataset is widely used in this context, more datasets are needed to demonstrate empirical utility and robustness of this approach. Furthermore, while the proprietary dataset used could be informative, without publishing or releasing this data, there is no means to assess its soundness, quality, etc.
* The paper writing would benefit from more attention; in particular, fleshing out the related work this is built upon (and perhaps reorganizing Section 4 into the beginning of the paper), clarifying the paper narrative throughout. Additionally, introducing the method presented with more focus would be helpful. The current structure is meandering and makes it difficult to follow—even if many of the pertinent details are included.

**Questions:**

* How does your approach handle immutable features?
* Suggest providing a simple, illustrative example for how your workflow would operate in the intended setting, from original user/state to resulting counterfactual explanation.

---

### Official Review · Reviewer_WF3J · 2025-10-31

**Soundness:** 1
**Presentation:** 2
**Contribution:** 1
**Rating:** 2
**Confidence:** 5

**Summary:**

This paper presents a conterfactual generation method aimed at enhancing customer retention in the context of e-commerce. According to the authors, the method should improve upon state-of-the-art methods like DiCE and GAN-based approaches, being one of the main advantages that it allows for specifiying which features are mutable (i.e. non-protected features). The method is based on converting tabular data in text form and generating embeddings via a encoder-only language model (BERT).

**Strengths:**

The paper is in general easy to follow and the structure of the paper is clearly specified and consequently followed.

**Weaknesses:**

This paper shows a number of significant weaknesses like the following:
- The paper fails to cite relevant previous work, especially the CERTIFAI framework: Shubham Sharma, Jette Henderson, and Joydeep Ghosh. 2020. CERTIFAI: A Common Framework to Provide Explanations and Analyse the Fairness and Robustness of Black-box Models. In Proceedings of the 2020 AAAI/ACM Conference on AI, Ethics, and Society (AIES ’20), February 7–8, 2020, New York, NY, USA. ACM, New York, NY, USA, 7 pages. https://doi.org/10.1145/ 3375627.3375812. Moreover, the CERTIFAI approach allows for speicifying protected features (conversely, mutable) which is one of the main novelties the authors assert.
- While the authors cite mutability as one of their main contributions, its implementation is not documented in the pseudocode. In fact, modifying any method to account for mutable and non-mutable features is almost trivial: in the authors case, as an additional condition in the inner for-loop. The authors should address how their method exactly implements mutability and why it should be treated as novel.
- The algorithm presented by the authors actually boils down to a greedy algorithm. It is difficult to see any novelty in such an algorithm.
- The authors claim their paper is centered around e-commerce, however there is no specific hint in the methodology or the evaluation to e-commerce apart from the closed-source dataset the authors use. In fact, the second dataset (adult) has nothing to do with e-commerce, raising concerns about the general framing of the paper. It is not clear why the authors chose adult as the second dataset and how it is relevant to the paper.
- There are no details on section 3 on the classification algorithms used, no hyperparameters were specified.
- The authors do not explain why proximity is more critical than diversity in the e-commerce domain (line157).
- The authors claim in Section 6 without justification that shorter L1 distance implies plausibility.
- Table 2 shows a much higher latency of the proposed algorithm compared with baselines. The authors should acknowledge this (strong) limitation.
- In Table 5 the authors claim differences in reconstruction error and L1 distance, but they only show coverage.
- The "ablation study" that the authors claim is not such a study, since the authors only perform a variation of one parameter (the embedding types) with only one mutable feature. In an ablation study, parts of the model are shut down to compare performance without it (hence the name ablation). The authors should change the name to "variation study".
- The authors exaggerate their contributions in Section 7, since the results do not back the conclusion that the method outperforms all existing methods in coverage, reconstruction error and L1 distance (see Table 2).
- The authors do not provide any source code.
- The citations are not properly formatted (use \citep instead of \cite).

**Questions:**

- Could you explain how the results in Table 4 correspond to an ablation study?
- Why did the authors choose to use the adult dataset when the paper is framed as e-commerce?

---

### Official Review · Reviewer_cMsZ · 2025-11-01

**Soundness:** 2
**Presentation:** 2
**Contribution:** 2
**Rating:** 2
**Confidence:** 3

**Summary:**

The paper proposes an embedding-based pipeline for counterfactual explanations tailored to e-commerce applications. Rows of tabular data are fist converted to short texts and embedded with BERT. The embeddings are then queried for _nearest unlike neighbors_. A greedy feature-editing search then modifies features one at a time to cross the classifier’s decision boundary.

**Strengths:**

* The proposed method is easy to understand and implement.
* The use of BERT can potentially bring the power of language modeling trained with huge amount of data into the tabular tasks.

**Weaknesses:**

* The experiments target at two very specific datasets: e-commerce and adult income. The title and abstract also suggest that the scope is very narrow. It is suspicious whether it can be used in other applications. More extensive experiments on data from different domain should be performed to prove its value.
* The proposed method is quite naive. I do not think using a BERT encoder to extract feature can be considered novelty.

**Questions:**

None

---

### Meta-Review · Area_Chair_VMen · 2026-01-05

**Summary:**

The paper addresses a relevant problem but raises consistent concerns of the reviewers regarding limited technical novelty, insufficient experimental depth and generality, and lack of methodological clarity and rigor. The approach is viewed as a straightforward application of BERT embeddings with nearest-neighbor search and a greedy heuristic. Evaluation is only on two datasets, and latency is worse than baselines. Writing quality needs to be improved and implementation details need to be provided. These concerns lead to a consensus toward rejection.

**Reviewer Concerns:**

- limited novelty

- Missing key prior work (e.g., CERTIFAI)

- Evaluation breadth and generality

- Latency disadvantages

- Missing methodological details and code

**Reviewer Scores:**

The author didn't provide rebuttal. I don't think the reviewers would have increased their scores.

---

### Decision · Program_Chairs · 2026-01-26

Reject